# The 3Rs Principle in Animal Experimentation: A Legal Review of the State of the Art in Europe and the Case in Italy

**DOI:** 10.3390/biotech10020009

**Published:** 2021-05-20

**Authors:** Enrico Maestri

**Affiliations:** Department of Law, University of Ferrara, 44121 Ferrara, Italy; mstnrc@unife.it

**Keywords:** 3Rs principle, legal enforcement, the case of Italy

## Abstract

The aim of this paper is to describe the essential points of Italian and European legislation governing the use of animals in biomedical experimentation. A close look will be taken at the principles of the 3Rs, which represent the mainstay of the legal architecture based on which a correct interpretation may be drawn of the legislative documents on animal experimentation. Furthermore, this paper will address the ways in which Directive 2010/63/EU is implemented in Italian legislation on the welfare of laboratory animals. In addition to an assessment of legal issues (such as the scope of jurisdiction of supervisory authorities tasked with issuing authorizations), it will include a discussion of cases of inadequate and insufficient implementation of the requirements laid down by Directive 2010/63/EU. Both the consistency of the interpretation of national legislation with the Directive and the direct effectiveness of the Directive in national law, in which animal testing has been and still is the subject of heated debate between supporters and opponents, will be examined.

## 1. Introduction

The creation of an adequate legal framework for the conduct of experiments on animals requires careful and responsible consideration of various interests at stake.

The conflicts among the obligation to maintain and improve human health, the valuable gains made possible by the freedom of science, the improvement of social welfare, the protection of the environment, and the deep concern about preventing animal pain and suffering cannot be resolved through general judgments. Animal protection legislation in Italy and in Europe thus calls for a careful examination and consideration of every single experiment on animals: this is a central element of the legislation [1].

The right to life is universally recognized, and this is explicitly stated in international treaties such as the Universal Declaration of Human Rights (Article 3), the Charter of Fundamental Rights of the European Union (Article 2), the International Covenant on Civil and Political Rights (Article 6), and the Convention on the Rights of the Child (Article 6). In Italy, the right to life and physical integrity is recognized by the Constitution (Article 2; Article 32). 

The binding principles regarding experiments on humans were laid down for the first time in history by the Nuremberg Code of 1947, which was the result of legal actions against physicians who were being tried in Nuremberg for the crimes against humanity perpetrated by the Nazi dictatorship. As part of the grounds for its judgment, the court set out the Nuremberg Code of Conduct for doctors [2].

On the basis of this document, a first draft was adopted by the World Medical Association in 1962 and was revised in 1964; the final version of the so-called Helsinki–Tokyo Declaration was revised and adopted by the 29th World Medical Assembly in Tokyo in 1975. The declaration, which would become binding worldwide, affirms that studies on human subjects are allowed only if the risks for the patient are minimized to the greatest possible extent: an assessment or minimization of risks for participants in clinical trials is possible only on the basis of broad scientific knowledge. The declaration further establishes that research may be conducted on human subjects only when all other scientific research options, including animal experimentation, have been exhausted [3] (p. 79). 

Freedom of scientific research is provided for in the Charter of Fundamental Rights of the European Union which states “The arts and scientific research are free” (Article 13). The freedom of science is an essential requirement for research and innovation to become an engine of development and well-being. In Italy, the Constitution recognizes the freedom of science as an absolute fundamental right (Article 33) and protects the quest for knowledge as an expression of human dignity. At the same time, the freedom of research is not absolute because, as stated in the European Charter for Researchers (2005), researchers are required to “adhere to recognized ethical practices and fundamental ethical principles”. On the other hand, the relationship between research and ethics raises complex questions, many of which are related to the specificity of the scientific field in which they operate. 

On this basis, experiments on animals are expressly provided for in legislation on pharmaceutical products and international guidelines applicable to pharmaceutical testing (OECD guidelines or European Pharmacopeia in the case of alternative methods of quality control for batches of vaccines). The possible benefit for patients receiving a new treatment and the burden on the animals used in animal experimentation must thus be balanced against each other, taking into account that, at present, there is no existing replacement for animals in basic and translational research or in the development of new therapies [4].

The relationship between humans and animals is more contradictory than ever and lies in a state of inner conflict between affection and personal interests. Up to now, animals have often been essential to research as a model organism, because experiments on animals provide important information about how drugs work and on the human toxicity of individual chemical products. This has given rise to a dilemma between health and safety requirements on the one hand and the moral need to protect laboratory animals on the other [5].

In this context, animal experimentation is subject to strict legal limits. Research projects are based—mandatorily—on compliance with the method of the 3Rs: replacement (to the extent possible, of trials that use animals with alternative research methods); reduction (in the number of animals sacrificed for research purposes); and refinement (the adoption of strategies that minimize the suffering of animals used in experiments). The objective of the 3Rs is to assure that greater care is taken to avoid making experimental tests intolerably stressful for animals and to use or develop alternative methods to in vivo experiments [6].

The aim of this paper is to describe the essential points of Italian, European, and supranational legislation governing the use of animals in biomedical experimentation. A close look will be taken at the principles of the 3Rs, which represent the mainstay of the legal architecture based on which a correct interpretation may be drawn of the legislative documents on animal experimentation. In particular, the main objective of this paper is to provide a legal description of the three pillars of animal research, i.e., the 3Rs—replacement, reduction and refinement—and the various definitions ascribed to them, from the original definition of Russell and Burch [7], to the current definitions provided by the guidelines and codes of practice of bioethics committees and by the most important scientific societies that focus on the care and use of experimental animals. Furthermore, the paper will address the ways in which Directive 2010/63/EU is implemented in Italian legislation on the welfare of laboratory animals. In addition to an assessment of legal issues (such as the scope of jurisdiction of supervisory authorities tasked with issuing authorizations), it will include a discussion of cases of inadequate and insufficient implementation of the requirements laid down by Directive 2010/63/EU. Both the consistency of the interpretation of national legislation with the Directive and the direct effectiveness of the Directive in national law, in which animal testing has been and still is the subject of heated debate between supporters and opponents [8], will be examined. Finally, an overview will be given of the main animal welfare guidelines, rules, and good experimental practices adopted in this field by scientific societies and bioethics committees that have attributed a pivotal role to the 3Rs method in the design of experiments that include animals.

## 2. The Principle of the 3Rs and Law: A Legal Genealogy

In general, the adjective “experimental” is used to refer to any method that enables a cause-and-effect relationship to be established between two phenomena/events on the basis of a hypothesis supported by already acquired knowledge and ultimately verified through experimental tests.

Experimentation should begin only if there are unfulfilled needs, i.e., when no good treatment exists or when the treatments currently used are poorly effective or cause many side collateral effects [9] (pp. 87–88).

A fundamental event marking the history of experimentation took place in the United States in 1937: a diethylene glycol (DEG) antifreeze caused the death of 107 people. An American pharmaceutical company had prepared a formula of sulfanilamide, a medicine used to treat streptococcal infections, using diethylene glycol (DEG) as a solvent. DEG was poisonous to humans, but this fact was unknown to the pharmaceutical company, which added raspberry flavoring and marketed the product as Elixir Sulfanilamide. Unfortunately, the formula caused mass poisoning, resulting, precisely, in the death of over one hundred people. After this tragedy, scientists administered the drug to some animals, which also died. The episode sent shockwaves through the scientific community at that time, and created the basis for a direct correlation between data originating from animal testing and their application in humans. In 1938, without questioning whether such experimentation could provide evidence that all animal species react in the same way to different chemical substances, the U.S. Congress approved a law (Food, Drug and Cosmetics Act) requiring pharmaceutical companies to test the safety of their products by conducting trials on animals. Since then, for the rest of the twentieth century, research was largely founded on in vivo testing, then considered a model of reference for reliable predictive analysis [10].

However, studies on animals should be avoided unless there is judged to be a real benefit for humans, and unless they are predictive for humans. Provided that animal protection laws are complied with and all possible forms of analgesia and anesthesia are used to spare animals from pointless cruelty, experimentation on animals can and must be ethically accepted [11]. Today, this acceptance is fundamentally based on precise justificatory grounds: a very widely held theoretical notion that has been defined as the “priority of the human being” [12]. According to this notion, animals are beings deserving of moral attention (moral patients), even though their interests are nonetheless deemed secondary compared to any competing interests of human beings (moral agents).

It is difficult to briefly outline the various moral reasons that may be put forward to justify such an attitude in favor of human beings and their interests; nonetheless, the fact remains that, alongside the priority accorded to human interests, there is also a widespread concern for animal welfare today [13]. Consequently, those who follow this position in respect of experimentation recommend minimizing the sufferance of animals within the framework of experimental practice. This goal may be pursued through the so-called principle of the 3Rs: replacement, reduction, and refinement.

The predictable pain suffered by animals during their use in experiments has induced public opinion and the scientific community to call for measures to ensure an adequate level of animal welfare [14]. The concerns about animal welfare, together with the growing use of animals in basic and applied research, led Russell and Burch, two British academics, to assess how decisions should be made in regard to the use of experimental animals while assuring their welfare. In their book entitled *The Principles of Humane Experimental Technique*, first published in 1959, Russell and Burch proposed the method of the three Rs: in the last fifty years, the three Rs have become widely accepted legal and ethical principles and now guide the conduct of animal-based scientific research in many countries in the world [15]. European animal (primary and secondary) legislation presently shows three trends: firstly, a constitutionalization of animal protection and welfare principles has taken place with the adoption of constitutional provisions (e.g., Article 20a of the German Constitution). This trend, also reflected in other national constitutions, increases animal welfare rights [16]. Secondly, a semantic clarification of the concept of animal has been imposed in civil law, which has called into question the traditional legal difference between animals and things originating from Roman law. Thirdly, the supranationalization of animal rights in the EU due to the transfer of legislative powers to the European Union in the sectors of agriculture and fishing, trade, the environment, and consumer protection. The EU itself has thus become the competent legislator here: for example, through the approximation of European legislation on animal experimentation with Directive 2010/63/EU.

The EU is authorized to adopt animal welfare regulations only to a limited degree: unlike environmental protection, animal welfare is not a goal of the EU and is therefore regulated by the Member States on the basis of national legislation. The EU can adopt animal welfare legislation which is binding for all the Member States if it prevents trade barriers and distortions of competition in the European single market. If the legislation on the keeping of farm animals and experimental animals differs in individual EU countries, traders based in countries with stricter laws and regulations would be disadvantaged due to the increase in production costs in the internal market. In order to counter such inequalities and disadvantages, the EU could adopt animal welfare regulations that are binding for all the Member States. However, not all types of animal experimentation are of “relevance for the internal market”, and those that are not cannot be regulated by the EU. If, on the other hand, the animal experiments are conducted in the framework of training and higher education or in basic university research according to the standards established by the EU, the issue is a relevant one and such experimentation is thus subject to regulation by the EU. In the “Treaty on the Functioning of the European Union” (TFEU, 2007), it is stated that “[i]n formulating and implementing the Union’s [...] policies, the Union and the Member States shall, since animals are sentient beings, pay full regard to the welfare requirements of animals, while respecting the legislative or administrative provisions and customs of the Member States” (Title II, Article 13 TFEU). Although animal welfare is not one of the common goals of the EU, it has the same relevance as other principles mentioned in Title II of the TFEU, such as guaranteeing an adequate protection of health or promoting gender equality.

Article 13 TFEU states:

“In formulating and implementing the Union’s agriculture, fisheries, transport, internal market, research and technological development and space policies, the Union and the Member States shall, since animals are sentient beings, pay full regard to the welfare requirements of animals, while respecting the legislative or administrative provisions and customs of the Member States relating in particular to religious rites, cultural traditions and regional heritage”.

On the one hand, animal welfare is recognized as having the same importance as the other principles mentioned in Title II of the TFEU, such as the promotion of gender equality, the guarantee of adequate social protection, the protection of health, the combating of discrimination, the promotion of sustainable development, consumer protection or the protection of personal data; on the other hand, animal protection is not an independent objective of the Union, but it has a significant impact on the policy sectors described. The concern for animals as sentient beings thus implies an institutional commitment to prevent them from suffering and recognition of their essential innate behaviors. It remains to be seen whether this improvement in the protection of animals, which is partially disputed in the legal literature, actually leads to consequences, in particular as regards the rulings of the Court of Justice of the European Union. The European Convention for the Protection of Vertebrate Animals Used for Experimental or Other Scientific Purposes (123/1986) was followed by Council Directive 86/609/EEC of 24 November 1986 on the approximation of laws, regulations, and administrative provisions of the Member States regarding the protection of animals used for experimental and other scientific purposes. This Directive, transposed into Italian law by Legislative Decree No. 116 of 27 January 1992, “on the protection of animals used for experimental purposes or for other scientific purposes”, gave rise to research and validation activities aimed at identifying new in vitro methods and modifying some existing in vivo methods to reduce the number of animals used and minimize their suffering and harm. Legislative Decree 116/1992 clearly adopted the principle whereby preference should be given to alternative methods, “as official methods, which involve the use of fewer and fewer animals as species and as categories” (Article 16b, Article 17b).

This approach was also reflected in Directive 2003/15/EC, which had set precise limits on the sale of cosmetics tested on animals.

Regulation (EC) No 1223 of 2009 calls for the gradual and—ultimately—total elimination of the possibility of performing tests on animals for cosmetic products in Europe. 

The new Regulation prohibits, in accordance with the 3Rs principle, animal testing of both finished products and ingredients or combinations of ingredients that will make up the finished product; the Regulation further prohibits the importation and marketing, in Europe, of products whose final formulation has been tested on animals. Moreover, because animal experiments cannot be completely replaced by an alternative method, it is necessary to specify whether the alternative method replaces the animal experiments partially or entirely (Commission Regulation (EC) No 440/2008 of 30 May 2008 laying down test methods pursuant to Regulation (EC) No 1907/2006 of the European Parliament and of the Council on the Registration, Evaluation, Authorization and Restriction of Chemicals (REACH)). Directive 2010/63/EU of the European Parliament and of the Council on the protection of animals used for scientific purposes, which came into force on 9 November 2010 and replaces the previously applicable Directive 86/609/EEC, promotes the standardization and harmonization of national legislation concerning the protection of animals in research, a subject already addressed by the former Directive. The national legislation of the Member States should aim to avoid, reduce, and improve the use of the animals for research purposes (cf. Article 4). Among the novelties of Directive 2010/63/EU, there is the inclusion of cephalopods, alongside vertebrates, as being worthy of attention (cf. Article 1, paragraph 3), as well as the broadening of the allowed purposes for carrying out experiments on animals, which include basic research, education and training, and forensic inquiries (cf. Article 5).

Like its predecessor, Directive 2010/63/EU establishes that Member States may, if necessary, adopt regulations and laws that go beyond the minimum standards defined by EU legislation; these may include, among other things, potential regulations on procedures for the approval of animal experiments or monitoring of public opinion on the issue of animal welfare (cf. Article 43, which regards the publication of non-technical summaries of authorized projects).

Under Directive 2010/63/EU (cf. Article 4, Articles 46–49), as well as the corresponding Convention of the Council of Europe (cf. Article 6), Member States of the European Union are called upon to seek and promote alternative methods to replace in vivo animal experimentation. To that end, organizations have been set up at the European and national levels, their task being to evaluate alternative and supplemental methods: avoiding unnecessary multiple tests should also help reduce the number of tests on animals. At a European level, the European Centre for the Validation of Alternative Methods (EURL ECVAM, European Union Reference Laboratory for alternatives to animal testing laboratory. Website: http://ecvam-sis.jcr.it/, accessed on 20 February 2021) provides information about well-established replacement approaches and validated supplementary and alternative methods.

## 3. The Principles of the 3Rs

The 3Rs have become a common point of reference for the EU Member States [17] and for a wide variety of organizations and committees whose goal is to avoid, to the extent possible, animal experimentation, or to improve conditions for laboratory animals; the principle is enshrined in the legislation of many EU countries [18] through Directive 2010/63/EU on animal experiments, which calls for the application of the “principle of replacement, reduction and refinement” (Article 4). 

Each Member State further has the obligation of identifying “single points of contact” (Article 47), which have the task of coordinating working groups at a national level and sharing their work with the respective health ministries and the EU Reference Laboratory for alternatives to animal testing (EURL-ECVAM), with the aim of facilitating contacts between the parties, shortening the time frame for the validation of alternative methods, and assuring a significant approach in the adoption of 3R strategies. The European Union Reference Laboratory for alternatives to animal testing (EURL-ECVAM) is located at the Joint Research Centre (JRC) of the European Commission. Moreover, the EURL-ECVAM specifies alternative methods to animal experimentation, in which the 3Rs principle is applied: an experimental procedure thus represents an alternative to animal testing only if it is capable of replacing animal testing, reducing the number of animals used, or refining the method to minimize the stress caused.

In 2005, the European Commission entered into a partnership with enterprises in various industrial sectors with the aim of developing alternatives to animal experimentation (European partnership for alternative approaches to animal testing (EPAA. https://ec.europa.eu/growth/sectors/chemicals/epaa_en, accessed on 20 February 2021)). The founding document of the EPAA is the so-called “3Rs Declaration”. 

Despite this strong effort to institutionalize the principles of the 3Rs, the latter have received different and conflicting interpretations. Dolan [19] has shown the heuristic insufficiency of the original definition formulated by Russell and Burch. Thus, over time, different meanings have been ascribed to the letter “R”. First of all, no one can deny the importance of respect in the treatment of laboratory animals. Other relevant Rs are: responsibility, because this word refers to an integral part of the management role of any project leader; reason, relevant for justifying the use of an animal in research; recognition, implying the most appropriate form of alternative which must be recognized and adopted; reflection, i.e., the need to reflect seriously on all the relevant literature in the search for suitable methods for implementing the three Rs; reconsideration, implying the feasibility of a new validated alternative, which will be taken seriously into consideration; and relief, whereby every means should be used to alleviate animal suffering.

The 3R principles have been incorporated into national and international legislation regulating the use of animals in scientific procedures with the aim of implementing more humane experimental approaches. With respect to laboratory animals, Russell and Burch considered replacement as the ultimate goal to be reached, whereas they considered reduction and refinement as two secondary objectives that could be more easily attained in the short term.

At present, the principles of the “three Rs” are increasingly being adopted as a basic framework for conducting high-quality scientific experiments and developing alternative tools for enhancing animal welfare [20].

Through the application of the 3Rs in the realm of animal experimentation for scientific research purposes, it is hoped to improve the understanding of the human–animal relationship; moreover, researchers can ask themselves whether “something” exists that separates us from animals and whether that “something” is sufficient to determine how we should treat them during an experiment. The legal reinforcement of the 3Rs principles sets an obligation for researchers, irrespective of their subjective will: they “have” an obligation to respect the 3Rs, even if they do not “feel” themselves to be obliged, and even though it is concretely difficult to respect them.

## 4. An Example of Legal Challenges in a European Member State: The Case of Italy

As was natural, given the highly controversial issue involved, the new Italian legislation has spurred much debate. Are the new rules better or worse than the previous ones?

According to the European Commission, in agreement with the Italian scientific community, Legislative Decree no. 29 of 2014 (which transposed the European Directive) established excessive limitations and violated European legislation, because it adopted overly strict criteria, following the entry into force of the Directive itself (cf. Table 1 below for details). In view of the incorrect transposition of Directive 2010/63, the European Commission initiated infringement proceedings in the European courts, demanding that Italy be fined.

However, a large part of the public viewed the new law as an improvement over the European Directive because it prohibited experiments on anthropomorphic monkeys (whereas the European Directive allowed exceptions), prohibited experiments for the production and control of war material, prohibited the reuse of animals in experiments causing a severe level of pain, and introduced more severe administrative and criminal penalties for those who violated the minimum standards established by law (cf. article 40).

The regulatory intervention of the Italian legislator aimed to guarantee a higher level of protection by establishing precise requirements and conditions for livestock farming activities and the supply and use of animals, with a view to progressively reducing their use and ultimately completely replacing animals with alternative practices and methods.

The 3Rs principle is explicitly recognized as a tool geared towards the implementation of alternative methods to the use of in vivo animal models.

In recital 6 of Directive 2010/63, it is stated that:

“New scientific knowledge is available in respect of factors influencing animal welfare as well as the capacity of animals to sense and express pain, suffering, distress and lasting harm. It is therefore necessary to improve the welfare of animals used in scientific procedures by raising the minimum standards for their protection in line with the latest scientific developments (6)”.

Therefore, it is necessary to take into account two conditions, laid down in recitals 10 and 11:

While it is desirable to replace the use of live animals in procedures by other methods not entailing the use of live animals, the use of live animals continues to be necessary to protect human and animal health and the environment. However, this Directive represents an important step towards achieving the final goal of full replacement of procedures on live animals for scientific and educational purposes as soon as it is scientifically possible to do so (10).

The care and use of live animals for scientific purposes is governed by internationally established principles of replacement, reduction and refinement (11).

Additionally, Article 1, paragraph 1, letter a of Legislative Decree 26/2014 affirms that:

“1. The present decree establishes measures relating to the protection of animals used for scientific or educational purposes; to this end, the following aspects are considered: (a) replacement, the reduction in the use of animals in procedures and the refinement in the techniques whereby animals are bred, housed, cared for and used in procedures”.

It should be highlighted that Article 2 of the Directive precludes the introduction of national measures that are stricter than those contained in the Directive itself. National measures providing for a higher level of protection than those in the Directive could be maintained, provided that they were already in force on 9 November 2010, but in such a case the Member States concerned would have to inform the Commission of such measures by 1 January 2013. The examination of Article 13 of Italian Law 96/2013—which delegated the implementation of the European Directive to the government—and its subsequent approval gave rise to a heated debate in the competent committees in the Chamber of Deputies and Senate, as well as within the scientific community, precisely because some of the rules of delegation established therein are more restrictive than those laid down in the European legislation. The legislative decree concerned, despite literally reproducing a large part of the Directive, introduced a discretionary part with a higher, more restrictive level of regulation compared to the European legislation. 

More specifically, the following provisions were introduced, which diverged from the European legislative provisions: prohibition of experiments and procedures that do not provide for anesthesia or analgesia, when they cause pain to the animal, except in cases where anesthetics or analgesics were being tested (Article 14, paragraph 1); prohibition against the production and control of war material; prohibition of toxicological tests with the lethal dose—LDSO and lethal concentration—LCSO protocols, except where they were required by national or international legislation or pharmacopeias, prohibition against the production of monoclonal antibodies through the induction of ascites, where other production methods exist and where it is not mandatory under national or international legislation or pharmacopeias; prohibition of research on xenotransplants, previously described as transplants of one or more organs from one animal species to another; prohibition of research on abused substances; prohibition of certain educational activities in primary and secondary schools and in university courses, with the exception of courses held in veterinary medicine faculties, and advanced university training for physicians and veterinarians (Article 5, paragraph 2); and prohibition against breeding, but not against using, dogs, cats and non-human primates intended for experimentation within the national territory (Article 10, paragraph 5).

It should be noted that the prohibition against breeding dogs, cats and non-human primates intended for experimentation within the national territory, as provided in Article 1 of the decree (scope of application), does not faithfully transpose the corresponding Article 1 of the Directive. It does not, in fact, transpose paragraph 2 of the latter article, which, in specifying the cases excluded from the provision, states as follows:

This Directive shall apply where animals are used or intended to be used in procedures, or bred specifically so that their organs or tissues may be used for scientific purposes.

It is worth highlighting that whereas Legislative Decree 116/1992 classified any use of animals liable to cause pain, suffering, distress or lasting harm as “experimentation”, the decree, like the Directive, excludes procedures from the scope of application which cause less pain than that caused by the introduction of a needle, thereby establishing a more objective threshold for the assessment of pain, which is at the same time one of the most complex aspects to quantify and one of the parameters mainly analyzed by the regulatory authority before issuing the necessary authorizations for carrying out experiments on animals.

It should also be stressed that the current decree did not faithfully and explicitly transpose Article 4 of Directive 63/2010 (Principle of replacement, reduction and refinement), although in many articles (e.g., Article 1, letter a, Article 13, Article 26, Article 31, and Article 37) reference is made to the 3Rs principle and experimental researchers are called on to implement it.

In any case, the Italian legislation failed to transpose Article 4 of the Directive, which reads:

1. Member States shall ensure that, wherever possible, a scientifically satisfactory method or testing strategy, not entailing the use of live animals, shall be used instead of a procedure. 2. Member States shall ensure that the number of animals used in projects is reduced to a minimum without compromising the objectives of the project. 3. Member States shall ensure refinement of breeding, accommodation and care, and of methods used in procedures, eliminating or reducing to the minimum any possible pain, suffering, distress or lasting harm to the animals. 

## 5. Conclusions

There is something essential behind all this debate: there is the presumption that, at some point in the future, the need to use animals in research will come to an end. However, as long as such use is still necessary, public acceptance is conditioned by the knowledge that the minimum number of animals will be used and that the minimum amount of pain will be caused to them, providing, in turn, the maximum benefit for humans, other non-human animals and the environment. The 3Rs provide a rational basis on which the use of animals can continue to receive public support.

However, and despite the undoubted advantages, animal testing should be tackled to minimize the harm and suffering inflicted on animals. Furthermore, if such damage or suffering occurs, an attempt should be made to produce it only when there is a proportionate and perfectly justified cause.

The ultimate and acceptable aim is unquestionable human well-being.

For all these reasons, today, the 3R’s principle plays a pivotal role in the ethics and law of animal experimentation.

A diligent implementation of the 3Rs, although not sufficient on its own, is therefore a necessary prerequisite for the justification of animal experiments. Irrespective of the degree of stress inflicted on animals, the study design must fulfil strict scientific quality requirements in terms of objectivity, validity, and repeatability. It is a fundamental responsibility of every researcher to ensure scientific quality in order to obtain the maximum scientific significance. Moreover, this goal must always be the central criterion of the scientific evaluation of research projects [21].

The 3Rs principle can fulfil its role only if an appropriate research plan is available and if the reduction principle is not misconstrued so that the number of animals is reduced to such a degree as to impair the study’s informational value. Should this occur, the principle of the 3Rs would be reduced to a magical formula whose legal enforcement would make it mandatory without researchers being able to understand, in their “internal legal culture” [22], the reasons why it must be observed.

## Figures and Tables

**Table 1 biotech-10-00009-t001:** European Directive 63/2010 vs. Italian Legislation 26/2014.

European Directive 63/2010		Italian Legislation 26/2014	Comparison
*Prohibitions*			
Article 2, par. 1 Stricter national measures 1. Member States may, while observing the general rules laid down in the TFEU, maintain provisions in force on 9 November 2010, aimed at ensuring more extensive protection of animals falling within the scope of this Directive than those contained in this Directive.		Art. 13 Delegated Legislation 96/2013. Criteria for delegation to the Government for the transposition of Directive 2010/63/EU of the European Parliament and of the Council of 22 September 2010 on the protection of animals used for scientific purposes. 1. In exercising the delegation for the implementation of Directive 2010/63/EU of the European Parliament and of the Council of 22 September 2010 on the protection of animals used for scientific purposes, the Government is required to follow, in addition to the principles and guiding criteria referred to in Article 1, paragraph 1, also the following principles and specific guiding criteria: [...]	The Italian delegated rules, subsequent to the entry into force of the Directive, provide for more restrictive measures than those set by European legislation
Article 2, par. 2 2. When acting pursuant to paragraph 1, a Member State shall not prohibit or impede the supply or use of animals bred or kept in another Member State in accordance with this Directive, nor shall it prohibit or impede the placing on the market of products developed with the use of such animals in accordance with this Directive.			Legislative gap
Article 14 Anaesthesia 1. Member States shall ensure that, unless it is inappropriate, procedures are carried out under general or local anaesthesia, and that analgesia or another appropriate method is used to ensure that pain, suffering and distress are kept to a minimum.		Article 14 Anaesthesia 1. Procedures that do not involve anaesthesia or analgesia are prohibited, if they cause intense pain following serious injury to the animal, with the exception of procedures for testing anaesthetics and analgesics.	The European Directive is not respected
		Article 5 Purposes of the procedures 2. Procedures cannot be authorized: (a) for the production and control of war material; (b) for toxicological tests with the protocols of the Lethal Dose -LD50 and of the Lethal Concentration -LC50, except in cases where it is mandatory by national or international legislation or pharmacopoeia; (c) for the production of monoclonal antibodies through the induction of ascites, if other corresponding production methods exist and are not mandatory by national or international legislation or pharmacopoeias; (d) for research on xenotransplants referred to in article 3, paragraph 1, letter q); (e) for research on drugs of abuse; (f) during the didactic exercises carried out in primary, secondary schools and university courses, with the exception of university training in veterinary medicine as well as higher university training for doctors and veterinarians.	The Italian legislation provides for restrictive measures not provided (or only partially) for by European legislation
Art. 1 Subject matter and scope 2. This Directive shall apply where animals are used or intended to be used in procedures, or bred specifically so that their organs or tissues may be used for scientific purposes.		Article 10 Animals used in procedures 5. The breeding of dogs, cats and non-human primates for the purposes referred to in this decree is prohibited.	Art. 10 par. 5 article 1 par. 2 of the European Directive is not faithfully implemented. The EU Directive provides for exceptions to experimental use on non-human primates.
*Alternative Approaches (3R)*			
Article 4 Principle of replacement, reduction and refinement 1. Member States shall ensure that, wherever possible, a scientifically satisfactory method or testing strategy, not entailing the use of live animals, shall be used instead of a procedure. 2. […] 3. […].		See art. 1 letter a, art. 13, art. 26, art. 31, and art. 37	The current legislation has not been transposed art. 4 of Directive 63/2010 in a corresponding and explicit mode
*OGM*			
*Article 3* Definitions		Article 10 Animals used in procedures 4. The breeding of genetically modified animals is permitted subject to an assessment of the relationship between harm and benefit, the actual need for handling, the possible impact it could have on animal welfare and the potential risks for human, animal health and for the environment.	The Italian legislation provides for more restrictive measures than those provided by the European Directive
For the purposes of this Directive the following definitions shall apply:			
1.	[…]. This includes any course of action intended, or liable, to result in the birth or hatching of an animal or the creation and maintenance of a genetically modified animal line in any such condition, but excludes the killing of animals solely for the use of their organs or tissues;		

## Data Availability

Not applicable.

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
