# Peer review of "The 3Rs Principle in Animal Experimentation: A Legal Review of the State of the Art in Europe and the Case in Italy"

_biotech, 2021, doi:10.3390/biotech10020009_

Round 1
Reviewer 1 Report
It is a very well-written legal manuscript with references and analysis of all legal documents which are related to the protection of animals used for scientific purposes. The provided information is not new. There are a lot of papers related exactly to the presentation and analysis of the Directives 86/609, 2010/63, REACH, etc. A major point that was noticed is that the author doesn't refer to the differences which exist between the European Directives issued by the European Union and the European Convention issued by the Council of Europe. Member States of the EU have an obligation to adopt the Directives into their national legal framework. On the other hand, the ratification of the European Convention is upon the willingness of the Member State of the Council of Europe. Although Chapter 4 of the manuscript is very interesting, in many cases the author expresses personal views which are not supported by related references. The provided table 1 is very interesting. The author provides personal interpretations of specific articles. The difficulty I had as a reader was that as I don't know all text of the Law, I can't understand its spirit, and of course, I can't easily accept or reject the interpretations given by the author.
I believe that this is a very good and very well-written legal document that couldn't be published in a journal like BioTech. I would recommend the author to send it in a journal related to legal issues and Law.
Author Response
See the revisions I applied to the attached file.

Reviewer 2 Report
Line 36 “right to life” why is this only human life? If you are arguing that others have lives too then it must be at least considered or outlined WHY the law concentrates on humans only.
75. You MUST address this issue: if animals are used in experimentation for human benefit, then it is assumed that they are similar; if they are similar then why are they not accorded the same ethical considerations?
If they are not similar then why are they used at all as it would be more to the point scientifically to use humans, even bearing in mind the guidelines for their use?
77 This reduction in the 3 R’s should also then be applicable to humans?
127 real benefit to humans why? This subject must be addressed rather than just assumed that humans are special if any progress in the ethics is to be made. What about the need to benefit animals, or the animals being tested at least?? In the age of vanishing species it is long overdue that this was at least considered in law ( and the environmental concerns might be the place here if they are already in European Law but no discussion of this).
135 There is plenty of evidence that “animals”, mammals at least who are principally used in experimentation are moral agents: that is know the difference between right and wrong. Again it is not very helpful sweeping all this away.
222 There is a real problem with “religious rights” re animal welfare as you know. Why if the animals are sentient is it permitted to kill them without pre-stunning IF you are of a certain religious belief? The religious belief of the human makes NO difference to the suffering of the animal, therefore there is an irrationality in this argument and if it is to continue this argument MUST be examined.
231 “essential innate behaviors”… what about as with humans, their learning and experiences? If they are sentient and learn voluntarily??
369 It is not the farming of animals that is the problem , it is necessary to have some farmed animals around for the environmental reasons of the EU, but it is HOW they are farmed. Assuming that all farm animals suffer is over stating the case and NOT helpful. The Italian public need further information perhaps?
498 How is it proposed to statistically measure “pain” or other emotions? Is this really relevant to the debate and if not then best left out.
General comments. I would have thought that the conclusions were already in the law's acceptance of the 3 R’s by the EU, so what is new about all this?
I commend the Italian efforts at further changes to benefit the animals, but this article does not strike me as being research so much as comment concerning EU legislation and how it works or does not on Animal Welfare. Is it really relevant to "Animals" Journal? I would have though that a short 2 page comment on development of the 3 R’s idea in Italian legislation would be sufficient?
The arguments above do however need addressing, and if the authors wished to do this it would add something new perhaps.
Author Response
See the revisions I applied to the attached file.
If you want to read the metaethical methodology that supports my attached legal paper, see the following arguments which I have not included in the text.
If you read this note below, you will find the answers to your bioethical questions that you indicated to me in the following page lines: 36, 75, 77, 127, 135, 222, 231.
Methodological Questions:
Law and Morals
When one speaks of bioethical issues related to animals, and when one feels a need to justify some form of legal legitimization of a subjective interest, it is important to distinguish between morals and law (Hart, 1961).
Distinguishing law from morals is essential in order to guarantee moral criticism and individual freedom (Hart, 1958).
The judgement of all individuals, based on which we decide what “the right thing to do” is, derives from the conscience of each one of us vis-à-vis that of others (we all answer to our own conscience): subjectivity becomes morality when it guarantees the existence of a number of conditions such as intersubjectivity, reciprocity, universalizability and impartiality (Rawls, 1971).
Some of course argue that morality is a question of taste and thus strictly personal (emotivism), but even if one were to admit an argument of this kind, it would be precluded from being used (or considered viable) in the public sphere due to its absolute controvertibility.
The law, by contrast, regards a jurisdiction that is external to every individual (each one of us is answerable to an impartial judge): it is not positivized morality, and thus does not coincide with the moral standards of the majority; on the contrary, its jurisdiction is heteronomous and has nothing to do with what is right, but rather with what is valid, i.e., what is lawful based on a decision of an external entity (parliament, court) that is entitled to produce, promulgate and apply law.
If law coincided with morality, we would end up with a totalitarian rule of the majority and there would no longer be any possibility of engaging in moral criticism against the law, given that law and morals would come to coincide (Mill, 1859).
Therefore, when we discuss bioethical questions in general, we must focus on the requirements intrinsic to law; a general and abstract nature first of all but – no less importantly – social pressure, effectiveness and a collective sense of compulsoriness and joint responsibility.
If we failed to do so, we would risk making a serious epistemological error, that of mistaking every one of our desires for a right, every one of our demands, interests or needs for a legal reason, wrongly identifying the “law that is” with the “law that should be” (Kramer, 1998).
Legal and Moral Rights
The classic distinction made in the language of statute law is between natural rights and positive rights. From common law systems we get the distinction between “moral rights” and “legal rights”, which is untranslatable into Italian and, what is worse, in a tradition in which law and morals are two clearly differentiated normative spheres, it is incomprehensible: in Italian, the expression “legal rights” sounds redundant and the expression “moral rights” sounds contradictory (Bobbio, 1995).
Consider the following two statements:
1) all animals are equal;
2) all human beings are born free and equal in dignity and rights.
The first sentence, taken from the book Animal Liberation by Peter Singer (1975), has a moral nature because it has no legal support as it is not recognized by any legal regulation.
Peter Singer claims that animals of all species have the right to equal consideration and respect. However, his claim does not express a legal right but a moral interest whose is hoped the protection. But the desired interests are not yet (legal) rights.
The second sentence, taken from article 1 of the Universal Declaration of Human Rights (1948), expresses a legal right, that is, a right whose authority derives from a specific mandatory legal regulation. The legal right is an exclusionary reason (Raz, 1999), that is, a self-sufficient reason whose validity and binding nature are not founded on morality.
Based on these methodological questions raised here, it is clear that an effective legal protection of animal welfare can be formulated only in terms of direct duties of people towards animals (Kant, 1775-1780). In other words, talking about people’s duties means talking about positive legal norms that prescribe behaviors for which animals are a direct point of reference. The attribution of subjective rights to non-human animals (Regan, 1983; Francione, 2008) is not something to be taken for granted (nor is it even for human animals); unless we take it to be an empty, generic concept, it can only mean this: individual animals cannot be used even for the benefit of their own species, much less that of natural ecology, or for the benefit of the human species/human animals. An ethic of animal rights is thus likely to work only in a Walt Disney movie.
Speciesism and legal enforcement of animal welfare
Placing speciesism and the two historical forms of intraspecies discrimination (Van De Veer, 1979), namely, racism and sexism, on the same plane is clearly questionable from a conceptual standpoint.
Whereas the concept of species rests upon an innate biological link, the notion of race has a cultural origin and arises from a belief present in historical social ideologies founded on the false premise that certain peoples are superior to others. One might object that a concept founded on a biological fact is likewise not sufficient to justify a given moral treatment. As a cultural qualifying element, it is not an ethically relevant fact, unless it overcomes, from a logical viewpoint, what Moore (1903) called the open-question argument; we can therefore reply to someone who makes reference to a biological fact in the same way. In this case as well we have to demonstrate why a biological fact or a natural tendency have relevance from a normative perspective. We have to ask ourselves whether the confines of species is actually an insignificant concept from an ethical viewpoint: can it really be dispensed with?
There is a difference between racism and speciesism that could help us understand to what extent the concept of species can serve as a basis for normative reasoning.
According to Midgely (1983, 106), this difference can be deduced from the following observation: “the human races are not significant groupings, animals species are. It is never true that in order to know how to treat a human being you have to know what race he belongs to. But in the case of animals, knowledge of the species is absolutely indispensable”. This affirmation can be called into question in many respects, but it contains a kernel of truth. From a scientific standpoint, the concept of species is a relative concept; the Platonic idea of the unchanging nature of species was completely debunked by Darwin’s theory. A scientific and taxonomic model of reference remains, however, as it cannot be dispensed with, and in any case the Darwinian principle of continuity between species remains compatible with species differentiation, as otherwise one should affirm, on evolutionary bases, that in reality there exists a single category comprising all living beings.
Certainly, Darwin rejected the notion of the eternal existence and eternal stability of species, but not in the same way as those who today affirm that species classification is a purely arbitrary social convention. Darwin denied the unchangeable historical form of the species, but Darwinian biology has never denied the reality of species as natural kinds. The human species, like all other species, is not eternal, but rather the contingent result of evolutionary history.
The notion of species can be considered a form of distinction; the notion of race, by contrast, is a form of discrimination. Not only: the equating of race with species has constituted the basis of racism and racism has been combated on a theoretical level precisely through the concept of species: “all men belong to the same species” (UDHR, 1948, Preamble). This is a clear example of the normative use of the concept of species.
If we go back to reflect on the epistemic value of the concept of species we can also understand the reason why this concept is relevant: it is not an ethical concept in itself, but rather expresses a factual epistemic indication, which is necessary for the formulation of a value judgment regarding an individual’s quality of life.
Biological species are in themselves expressions of normality, normal forms of living beings. They are results of a division into ecological niches, which ensure the fulfilment of certain expectations and, accordingly, the preservation of a species. The particular aspiration of each species is connected to a certain expectant attitude, which considers the occurrence of what is expected to be normal, and the non-occurrence thereof as a deviation, an anomaly. No life exists, not even human life, without such a normality that gives rise to an expectation.
Without the concept of species it would be impossible to justify the ideal of equality among species, because if only “acosmic persons” (Spaemann, 1996/2017) existed, we would be able to affirm neither their equality nor their inequality. The species acts as a biological normality, on the basis on which a moral judgment may be constructed, since all are equal within the same category. However, this should not lead us to think that the normal-normative distinction (which the concept of species allows us to make) implies the ontological and ethical superiority of human beings; on the contrary, it allows us to affirm the value of diversity, to avoid the alternative between a humanized animal and a reified one, to attribute value to all living beings, human animals and non-human animals.
However, if the species distinction has no moral relevance, how should we take into account the differences among beings of different species? In this regard, Midgley (104) observes: “Is being cannibals the same as being carnivores, or is there a significant difference? [...] How should the problem of priority be resolved?”
In the view of Singer (1979), founder of the animal liberation movement, biological life (human and animal) says nothing about our moral capacities: these do not derive from a biological basis that has gradually evolved and become differentiated. However, Singer’s argument, consistent with the dualism of the quality of life, contradicts the naturalist assumption of the biological non-discontinuity between one species and another. If our consideration of the moral weight of each individual derives from a process of grading the capacity to have interests, we will not only have a qualitative distinction between superior and inferior sentient beings, but we will also have further distinctions in relation to the mental states that Singer might identify in the biological nature of all individuals that possess a central nervous system.
According to Weiss (1978), attributing moral subjectivity based on the recognition of certain mental states or certain cognitive skills has revealed to be a flawed way of addressing the problem of the relationship between humans and animals: one repeats the same mistake as when attempting to attribute moral status by identifying characteristics that an individual being, not necessarily a human or a non-human animal, must have. How can a subjective perception (for example, self-awareness or rationality) be used to construct an objective hierarchy of values on the basis of which to assign the status of moral agent or moral patient to an individual? How is it possible to objectively attribute or deny a sensation that only we ourselves can feel?
The risk once again is that of likening animals to humans. The underlying assumption of such reasoning, whereby one seeks to establish the relationship between humans and animals on moral grounds, is based on the anthropomorphic conviction that if animals are recognized to have some form of thought or conscience, it will be like our form of thought or conscience, albeit less developed. This theory risks generates a process whereby animals come to likened to humans, resulting in a form of reverse speciesism or, worse, a form of misanthropic anti-humanism. Consequently, speciesism is not a form of discrimination and cannot be expected to have a significant role in legal reasoning.

Reviewer 3 Report
Animal experiments are indispensable in biomedical research. This manuscript reviews the legislation of animal experiments in Europe, introduces the origin of 3R and how to protect laboratory animal welfare in animal experiments. This manuscript is useful for readers for a better understanding of 3Rs theory and promotion of biomedical study.
The manuscript should focus on showing the importance of 3Rs theory and how the Directive 2010/63/EU reflects the spirit of 3Rs. It is recommended to briefly clarify the legal system of animal experimentation in Europe without mentioning too much content that is not relevant to 3Rs.
Author Response
I have taken your suggestions seriously and as a result I have completely deleted paragraph 5 dedicated to describing the current institutional status of the scientific and ethical committees dealing with experimentation and animal welfare.

Reviewer 4 Report
The authors present an important review of Italian, European and supranational legislation governing the use of animals in biomedical experimentation. This paper is especially timely and should be of particular interest with animals playing a pivotal role in the rapid development of COVID-19 therapies, vaccine candidates, and understanding key features of the virus’s biology.
My comments to the authors to address are as follows:
Section 1 Introduction
Ln 25, “the obligation to maintain and improve human health” is quite limited in the perspective of how animals are used in research, even constrained to only medical research. Generally this should be viewed as a societal benefit (which might include the environment, other animals, etc). For the following points, the authors are generally using human patients and medicines as the example, but it should be acknowledged from the ethical perspective there is more but they are specifically considering only human medical benefit.
Ln 69, “an area of tension between affection and personal interests” is a curious statement – are the authors indirectly implying that one cannot like animals if they also use them in experiments? It is an important transitional statement after describing the ways in which laws compel the use of animals which are generally based on a utilitarianist viewpoint.
Ln 78, the 3Rs is a framework, the objective of each of the Rs should be briefly articulated from the contemporary perspective.
Section 2 3Rs and Law (please review numbering of main and subsections throughout the manuscript).
Para 110, excellent example of one of the drivers for legislation.
Ln 127, Yes, nicely presented.
Para 146, This is confusing “the predictable potential pain” please revisit this, essentially there are inherent harms that exist on a spectrum related to the research. The comment that the 3Rs have been accepted as legal and ethical principles seems incorrect, they have been adopted into the regulatory framework but do not serve as an ethical principle independently.
290 Section 3 The 3Rs method. This section can be shortened to be much briefer with appropriate referencing since the focus of this manuscript is in the legal manifestations versus the theory itself. An example of this is paragraph starting at 329, it is not clear how the authors expansion on the R’s terminology is adding value to the message as each of these concepts are woven into more modern interpretations of the 3Rs.
Section 4
Table 1 is an excellent summary of the interpretation of the directive
Section 5
Ln 560, “ethics committees focused on animal experimentation still remain an isolated phenomenon: they are mostly separate entities. They are independent bodies that are not hierarchically subordinate to anyone, and precisely for this reason they can decide whether to add to their institutional aims research interests they consider to be consistent with their mandate.” This statement is unclear to me what the authors hope to convey about the role of an ethics committee that would transition into why it is valuable to have a networked approach in subsequent paragraphs. Please reconsider this and clarify with more concise language.
Section 6, Conclusion
Para 612, this is a concise summary that is well done. The conclusion is thoughtfully structured, and this same compact approach should be considered throughout the manuscript to balance the inclusion of quite a lot of legal text and reduce the density to the readers.
Congratulate the authors on a highly informative manuscript that will be of great interest and relevance to the field.
Author Response

(The authors gave the same response as above.)

Round 2
Reviewer 1 Report
This is a well-written manuscript. Following the previous revision, the author justifies his choice to submit this manuscript to the journal BioTech and this is acceptable. Chapter 4 of the manuscript is also very informative and well structured. Chapters 2 and 3 have to be reconsidered.
More specifically:
1) In chapter 2 the sub-titles are irrelevant to the texts. Thus, the author has to rearrange the structure (title, sub-titles) at this part of the manuscript.
- In "2.1. International and EU legal framework" nothing is said about international legislation
- In "2.2. Council of Europe" is considered European level and not International (as it is written)
- In "2.3. European Union" why is separated from 2.1. EU legal framework
- In "2.6. Directive 2010/63/EU" why is separated from the previous sub-chapters related to the EU.
2) In chapter 3. "3. The 3Rs method" is not supportive of the text which is written below and we are proposing to change it. What "method" means?
3) The first part of the title of the manuscript "The 3Rs Principle in Animal Experimentation" could be better addressed in the manuscript.
Reviewer 2 Report
This ms is improved but I still feel it can be cut considerably for this journal, which is not likely to have many lawyers as readers. It is too long winded.
Round 3
Reviewer 1 Report
The revised manuscript is now accepted for publication